# Engraftment of Allotransplanted Tumor Cells in Adult *rag2* Mutant *Xenopus tropicalis*

**DOI:** 10.3390/cancers14194560

**Published:** 2022-09-20

**Authors:** Dieter Tulkens, Dionysia Dimitrakopoulou, Marthe Boelens, Tom Van Nieuwenhuysen, Suzan Demuynck, Wendy Toussaint, David Creytens, Pieter Van Vlierberghe, Kris Vleminckx

**Affiliations:** 1Department of Biomedical Molecular Biology, Ghent University, 9000 Ghent, Belgium; 2Cancer Research Institute Ghent (CRIG), 9000 Ghent, Belgium; 3Department of Internal Medicine and Pediatrics, Ghent University, 9000 Ghent, Belgium; 4Laboratory of Mucosal Immunology and Immunoregulation, VIB-UGent Center for inflammation Research, 9000 Ghent, Belgium; 5Department of Pathology, Ghent University Hospital, 9000 Ghent, Belgium; 6Department of Biomolecular Medicine, Ghent University, 9000 Ghent, Belgium; 7Center for Medical Genetics, Ghent University, 9000 Ghent, Belgium

**Keywords:** *Xenopus*, tumor transplantations, genetic cancer model, *rag2* knockout, immunocompromised

## Abstract

**Simple Summary:**

While the mouse is without doubt the most studied animal for experimental cancer research, aquatic vertebrates such as zebrafish have also contributed to the field. More recently, thanks to the Nobel-prize winning technology of CRISPR/Cas mediated genomic engineering, the frog *Xenopus tropicalis* has emerged as an additional powerful model for studying human cancer. Via CRISPR-mediated genome editing, several models for different human cancers have been obtained in this animal. However, what has been lacking in *Xenopus* is the possibility to transplant tumor cells between different frogs. This is important to allow better characterization of the tumor cells and exploration of therapeutic opportunities. In this paper, we describe the generation of a genetic mutant in *Xenopus* *tropicalis* that has a compromised immune system, thereby allowing the grafting and expansion of tumors obtained in this species. In addition, an optimized protocol is provided for the irradiation of wild-type *Xenopus* frogs that subsequently are temporarily immunocompromised and during that period allow tumor engraftment. This work will expand the toolbox for modeling human cancer in *Xenopus tropicalis*, thereby further establishing it as a powerful experimental cancer model.

**Abstract:**

Modeling human genetic diseases and cancer in lab animals has been greatly aided by the emergence of genetic engineering tools such as TALENs and CRISPR/Cas9. We have previously demonstrated the ease with which genetically engineered *Xenopus* models (GEXM) can be generated via injection of early embryos with Cas9 recombinant protein loaded with sgRNAs targeting single or multiple tumor suppressor genes. What has been lacking so far is the possibility to propagate and characterize the induced cancers via transplantation. Here, we describe the generation of a *rag2* knockout line in *Xenopus tropicalis* that is deficient in functional T and B cells. This line was validated by means of allografting experiments with primary *tp53*^−/−^ and *apc^+/−^/tp53^−/−^* donor tumors. In addition, we optimized available protocols for the sub-lethal irradiation of wild-type *X. tropicalis* froglets. Irradiated animals also allowed the stable, albeit transient, engraftment of transplanted *X. tropicalis* tumor cells. The novel *rag2*^−/−^ line and the irradiated wild-type froglets will further expand the experimental toolbox in the diploid amphibian *X. tropicalis* and help to establish it as a versatile and relevant model for exploring human cancer.

## 1. Introduction

Tumor transplantation has been recognized as an indispensable tool in the cancer research field and has been successfully performed not only in mammalian species such as mice, reviewed by Sharkey and Fogh [1], but also in non-mammalian vertebrates such as zebrafish, reviewed by Gansner et al., [2]. Cancer immunoediting and, more specifically, cancer immunosurveillance are important processes that can severely hamper the engraftment of tumors in immunocompetent hosts [3]. In order to escape this phenomenon, either inbred or immunodeficient animals are required, thus allowing stable tumor progression after transplantation. Researchers working with mice were able to generate, amongst others, the ‘nude mice’ (lacking the thymus and thus functional T cells), the NOD-SCID and SCID-beige mice that are deficient in both the T and B cell pool, and finally the NSG or NOG mice that additionally lack functional NK cells [4]. More recently, zebrafish have joined the field. Several protocols and resources are available in this species to achieve the stable engraftment of transplanted cells such as, for example, sub-lethal irradiation [5], the use of *rag2^E450fs^* immunocompromised animals [6] and the use of syngeneic zebrafish lines, e.g., the CG1-strain [7]. Furthermore, for xenograft experiments, this species holds great promise as the transparent *casper* strain allowed the tracing and functional characterization of fluorescently labeled human tumor cells [8]. Most recently, the Langenau lab generated adult *prkdc*^−/−^, *il2rgα*^−/−^ immunocompromised zebrafish in the *casper* strain that allowed the robust engraftment of human cancer cells [9].

*Xenopus*, like the zebrafish, enjoys transparency in the embryonic stages, allowing the tracing of fluorescently labeled cells. Furthermore, the *Xenopus’* innate and adaptive immune cells and mechanisms show high conservation with their respective mammalian counterparts [10]. Despite the emergence of *Xenopus tropicalis* as a cancer model, thanks to the ease with which genetically engineered *Xenopus* models (GEXM) can be generated [11], so far experiments with tumor transplantations have not been documented for this species. Transplantations of *X. laevis ff-2* lymphoid tumor cells in MHC homozygous partially inbred adults of the *X. laevis ff* strain have led to the interesting finding that grafts are accepted in transplanted tadpoles but rejection occurs in transplanted adults [12,13]. This phenomenon is believed to be due to the second histogenesis present in the thymus during and after metamorphosis [12,13]. Recently, Rollins-Smith and Robert [14] described a protocol to induce lymphocyte deficiency by subjecting *X. laevis* frogs to sub-lethal gamma irradiation. Another study [15], showed engraftment successes after transplanting the 15/0 lymphoid tumor cell line (from a spontaneous *X. laevis* thymoma) in these *X. laevis* irradiated hosts. We describe here the generation and validation of a novel immunodeficient *rag2^−/−^ X. tropicalis* line that is suitable for allotransplantation experiments. Furthermore, we optimized and validated protocols for transplanting primary *Xenopus* tumors in irradiated *X. tropicalis* hosts. We believe these robust tools will be of high value for *Xenopus* tumor transplantation experiments and tumor immunity studies in general.

## 2. Results

### 2.1. Generation of a rag2^−/−^ Line

In order to generate a *Xenopus tropicalis rag2^−/−^* line, an sgRNA was designed, targeting the first fifth of the *rag2* single exon gene. Wild-type embryos were injected with a mixture of the selected sgRNA and Cas9 recombinant protein (Figure 1A). To analyze the editing efficiency, stage NF 41 embryos were lysed and genotyped. Amplicon deep sequencing of the targeted region in the *rag2* gene revealed a major proportion of reads showing a specific 4 bp deletion, which is in agreement with the inDelphi CRISPR repair outcome prediction algorithm [16]. Correlation analysis revealed a significant high overall correlation between predicted and experimentally observed frequencies of variant calls (Pearson r = 0.9886, *p* < 0.0001) (Figure 1B), confirming previous findings proposing inDelphi as a suitable method for predicting CRISPR/Cas9-induced repair outcomes in *X. tropicalis* [17]. For obtaining homozygotes (see schematic Figure 1A), first, crispant mosaic mutant animals were raised until adulthood, outcrossed with wild-type animals and checked for germline transmission in their progeny. Heterozygote *rag2^+/mut^* animals were subsequently intercrossed and homozygote *rag2^mut/mut^* animals were selected using a mixed Heteroduplex Mobility Assay (mHMA) genotyping technique [18] (Figure 1C top; Appendix A). Sanger sequencing confirmed the presence of a biallelic 4 bp deletion in the homozygous mutant animals (Figure 1C bottom). This deletion induces a frameshift after amino acid 91 resulting in a non-functional protein. Therefore, these animals are further referred to as *rag2^−/−^*.

### 2.2. Transplantation of X. tropicalis Tumors in the X. tropicalis rag2^−/−^ Line

To assess transplantation potential in the novel *rag2*^−/−^ line, a thymic tumor originating from an adult *tp53*^−/−^ animal from a previous study [19] was isolated (Figure 2A). Two parallel transplantations were performed: 5 × 10^6^ tumor cells were transplanted intraperitoneally (IP) in a *rag2*^−/−^ and a wild-type adult as illustrated in Figure 2B. Ten weeks post-transplantation, the *rag2*^−/−^ animal showed obvious signs of lethargy, while the transplanted wild-type showed no signs of discomfort. A clear externally visible outgrowth was present in the *rag2*^−/−^ animal close to the transplantation injection site (Figure 2C). Upon dissection, multiple sites of engraftment were observed on the abdominal muscle wall and in the peritoneal cavity (Figure 2D). Histopathological analysis of the tumors revealed the presence of both epithelial and mesenchymal cell clusters, thereby showing morphological similarities to the donor tumor (Figure 2E, top). Interestingly, multiple zones with neovascularization were present in these tumor engraftment sites (Figure 2E, top). In addition, immunohistochemistry showed high proliferative capacity in both donor and engrafted tumors, as indicated by PCNA immunostaining (Figure 2E, bottom). Finally, the mixed HMA method confirmed the inclusion of the same *tp53* mutational variant, present in both the donor and the engrafted tumor (Figure 2F; Appendix A). In addition, for a transplantation validation experiment (Figure 2G), a donor tumor aliquot (3.3 × 10^6^ tumor cells) was frozen at −80 °C in serum and, after nine months, was gently thawed for transplantation in another *rag2^−/−^* froglet. Already 43 days post-transplantation, the animal showed indications of disease and was dissected. Obvious signs of engraftment were already macroscopically present in the kidney and liver, further confirmed after histopathological assessment (Figure 2G).

Furthermore, we wanted to show the transplantation potential for another tumor type. We harvested tumor ascites cells from an *apc^+/−^/tp53^−/−^* adult frog harboring a severe liver tumor and transplanted 2 × 10^7^ tumor IP ascites cells in a *rag2*^−/−^ froglet (Figure 2H). Twenty-seven days post-transplantation, the host animal showed symptoms of disease and was subjected to analysis. Upon dissection, signs of engraftment were present in both the liver and the lungs, and the kidney showed a pale appearance. Histological analysis revealed multiple engraftment regions in the host liver, with clear histological similarities to the original liver tumor in the donor animal. Furthermore, sequencing of a whole liver piece (including the graft) of the host animal revealed the presence of a unique 1 bp deletion in the *apc* target region and two unique 4 bp deletions in the *tp53* target region, being exactly the same mutations carried by the liver tumor in the donor animal, thereby confirming the non-host origin of these liver grafts (Figure 2H). Of note, the low percentage of the INDELs in the *apc* and *tp53* genes (~5%) compared with the original tumor is due to the fact that a piece of liver was sampled that still contained many normal liver cells next to the mutant tumor cell infiltrations. It is also worth mentioning that the liver is a highly perfused organ and, in *Xenopus,* the red blood cells retain their nucleus. All together, these data show that adult *rag2*^−/−^
*X. tropicalis* animals allow stable allografting of transplanted GEXM-derived tumor cells in different cancer contexts. 

### 2.3. Transplantation Validation in Irradiated X. tropicalis Animals

Efficient tumor cell transplantation might also be achieved via alternative techniques apart from the generation of the *rag2*^−/−^ line. Immunocompromised *X. laevis* animals can be obtained by sub-lethal gamma irradiation [14]. In order to generate irradiated hosts in *X. tropicalis*, the optimal dose suitable for the successful allografting of tumor cells needed to be determined. We irradiated (X-rays) three different groups of 4-month-old *X. tropicalis* froglets (8 Gy (n = 3), 10 Gy (n = 3) and 12 Gy (n = 3)). Approximately one week post-irradiation, all cohorts were euthanized and dissected. Major lymphoid organs (spleen and liver) and peripheral blood were checked to address irradiation potential. Natt and Herrick peripheral whole blood staining revealed significant reduction in white blood cell (WBC)/red blood cell (RBC) ratios in irradiated animals compared with the non-irradiated siblings (*p* = 0.0012) (Figure 3A). Of note, no significant differences were present between the three irradiated groups. Quantification of CD3 immunohistochemical stainings revealed that irradiation majorly impacted T cell levels in both spleens and livers (Figure 3B,C). For spleens, compared with the non-irradiated controls (51.9% ± 4.5), irradiation with an 8 Gy dose already induced a significant decrease in CD3 positivity (36.0% ± 5.9, *p* < 0.05). This effect became more pronounced when irradiating to 10 Gy (15.7% ± 2.1, *p* < 0.001) and to 12 Gy (4.9% ± 1.9, *p* < 0.0001). Additionally, in the livers, a similar dose–ratio trend was observed (non-irradiated (4.0% ± 1.8), 8 Gy (1.5% ± 0.5, *p* = 0.08), 10 Gy (0.4% ± 0.1, *p* < 0.05) and 12 Gy (0.2% ± 0.1, *p* < 0.05)). We therefore propose that irradiation up to a dose of 12 Gy is preferred for the optimal reduction of T cell numbers, thereby displaying the highest potential for successful tumor transplantation applications.

In parallel with the previously mentioned experiment in *rag2^−/−^* animals, we validated the transplantation potential of *tp53*-mutant GEXM tumor cells also in an irradiated animal. For this purpose, an irradiated froglet (12 Gy) and a non-irradiated sibling were injected intraperitoneally with 1 × 10^7^ live tumor cells. To avoid any risk of repopulation of functional immune cells after the irradiation procedure, the froglets were analyzed already 3 weeks post-transplantation, in absence of any external signs indicative for engraftment. Nevertheless, a clear increase of tumor cells circulating in the peritoneal cavity was observed in the irradiated transplant (non-RBC/RBC = 0.67 ± 0.057) compared with the non-irradiated transplanted control (non-RBC/RBC = 0.14 ± 0.023). Natt and Herrick staining showed that the non-RBC fraction in the irradiated transplant primarily represented tumor blast cells (Figure 4A). Furthermore, in-depth histological analysis revealed tumor engraftment in both the kidney and the liver of the irradiated transplanted animal, whereas the non-irradiated transplanted control did not show any signs of engraftment (Figure 4B,C). Similar to what was found for the *rag2*^−/−^ animal, tumor grafts observed in the irradiated transplant also showed high proliferative capacity as indicated by PCNA immunostaining (Figure 4B,C).

## 3. Discussion

Donor cell rejection by the host organism after (allo)transplantation is a common hurdle, jeopardizing the *bona fide* assessment of the engraftment potential of tumor cells. In absence of syngeneic models, the availability of immunocompromised animals is an absolute need to show evidence of engraftment after transplantation and to allow further phenotypic analysis of cancerous cells. 

We describe the generation of a novel *X. tropicalis rag2^−/−^* line as a useful tool for transplantation experiments. Due to the central role of the Rag2 protein in the process of V(D)J recombination, these animals should lack mature T and B cells. Similar to what has been shown in zebrafish [6], the *X. tropicalis rag2^−/−^* animals used in this study also allowed allografting of primary tumor donor cells injected intraperitoneally. Especially for longer incubations and serial tumor transplantations, this line is recommended over irradiated animals, where the transient nature of the immunosuppression might eventually hamper stable engraftment. In a first experiment, 10 weeks post-transplantation, solid tumor grafts were visible at the injection site in the *rag2^−/−^* animal, whereas no signs of engraftment were observed in the control animal. In a second experiment, we could confirm the application potential of the *rag2*^−/−^ line via the transplantation, with cells resulting from a (genetically) different tumor already showing engraftment after 27 days. Of note, previous transplantation studies with lymphosarcoma cells in *X. laevis* have shown how infectious mycobacteria-induced granulomas were mistakenly interpreted as the engrafted tumor cells [21,22]. Therefore, we would like to state that the validation of engraftment should not be based solely on histological assessment. In our study, for example, assessment of the engraftment was performed via endpoint histopathological analysis with an additional genotypic validation. 

Next to mutant or genetically modified hosts, the use of irradiated zebrafish [8] and mice [23] has assisted greatly in the field of cancer research. For *Xenopus tropicalis,* no data is available showing the potential of using this technique for performing allotransplantations. We showed that irradiating froglets with a dose of 12 Gy reduced T cell numbers approximately 10-fold in the spleen and 20-fold in the liver. We furthermore showed that this dose allowed efficient engraftment of *tp53*^−/−^ tumor cells 3 weeks post-intraperitoneal injection. Of note, using lower doses of irradiation might also be sufficient to allow the engraftment of host tumor cells. Goyos and colleagues [24] showed that a 10 Gy irradiation dose already induced an inhibitory effect on thymocyte survival in *X. tropicalis.*


We hypothesize that engraftment success depends on multiple parameters such as tumor type, injected cell numbers, injection site and incubation time in the host. Regarding the latter, it is known that the repopulation of functional immune cells in irradiated animals can impair stable engraftment of tumor cells. In zebrafish, the repopulation of myeloid, lymphoid and immune precursor cells is observed already 2 weeks after irradiating adult zebrafish with 12 Gy [5]. Considering this caveat, the availability of the *rag2^−/−^* line offers more flexibility with higher engraftment success rates even for long-term experiments. 

Taken together, both the *rag2* knockout animal and the protocol for irradiating wild-type froglets add a new and important tool to strengthen the application of cancer modeling in the diploid amphibian *Xenopus tropicalis*. We are convinced that with this novel *rag2*^−/−^ line—and the ease with which irradiation can be performed—studies on immune surveillance and tumor immunity will be significantly aided. 

## 4. Material and Methods

### 4.1. CRISPR/Cas9 Mediated Generation of Mosaic Mutant X. tropicalis Animals

The CRISPRScan software package [25] was used for the design of the *rag2* CRISPR sgRNA. A 5′-gaattaatacgactcactataggGTCTTCCCTCCATGAATGgttttagagctagaaatagc-3′ oligo along with the reverse oligo: 5′-aaaagcaccgactcggtgccactttttcaagttgataacggactagccttattttaacttgctatttctagctctaaaac-3′ was purchased (Integrated DNA Technologies, Coralville, IA, USA). First, DNA was prepared by the annealing of the two primers and by PCR amplification. The DNA template was in vitro transcribed using the HiScribe™ T7 High Yield RNA Synthesis Kit (New England Biolabs, Ipswich, MA, USA). The sgRNA was subsequently isolated using the phenol-chloroform extraction/NH_4_OAc precipitation method [26]. RNA quantity was calculated by Qubit^®^ 2.0 Fluorometer (Thermo Fisher Scientific, Waltham, MA, USA) measurement and quality was visually confirmed by agarose gel electrophoresis. A detailed guideline for generating the NLS-Cas9-NLS protein can be found in our previously published work [27]. After setting up natural matings, 2-cell stage embryos were injected unilaterally with a 1 nl pre-incubated (30 s at 37 °C) mix of sgRNA and Cas9 protein. Gene editing efficiencies were evaluated quantitatively by targeted amplicon next-generation sequencing (as described below). The InDelphi in silico prediction algorithm was included to validate endogenously observed frequencies of variant calls [16].

### 4.2. DNA Extraction and Sequencing

Gene editing was assessed by subjecting PCR-amplified sgRNA targeted regions to deep sequencing followed by BATCH-GE analysis [28]. DNA, from either whole embryos (three embryo pools each containing three stage NF 41 embryos) or from dissected tumors, was extracted via an overnight incubation (55 °C) in DNA lysis buffer (50 mM Tris pH 8.8, 1 mM EDTA, 0.5% Tween-20, 200 μg/mL proteinase K), followed by a 5 min boiling step and a final centrifugation. Primers used in this study for amplification were: *rag2^fw^* 5′-GCTATCTGCCTCCACTTAGAC-3′ and *rag2^rv^* 5′-AATGTCAATGGTGTCATCATC-3′ with an extra internal primer used for Sanger sequencing *rag2^int^* 5′-TCTCCTATTGACTGAAGATGCC-3′, *tp53^fw^* 5′-CAGTGCTTATTGTTACCTCCA-3′, *tp53^rv^* 5′-CATGGGAACTGTAGTCTATCAC-3′, *apc^fw^* 5′-CATCCTAACTCTGCCCAA-3′ and *apc^rv^* 5′-ATAATGTTCTGGTGGGCT-3′. The methodology for Sanger sequencing and correlation analysis between in vivo versus in vitro CRISPR mutational repair outcome can be found in [17].

### 4.3. (Mixed) HMA Genotyping Method

For genotyping the *rag2* line and the tumor (graft) cells, WT DNA (i.e., DNA from non-injected frogs) was amplified in parallel with each unknown DNA sample via a standard PCR. Subsequently, equal quantities of both products—PCR-amplified WT and unknown sample DNA—were mixed and eventually subjected to HMA in parallel with all the unknown samples individually (unmixed). This was completed by incubation of the samples at 98 °C for 5 min, followed by a 4 °C holding temperature incubation using a transition with a ramp rate of 1 °C/s. Finally, the PCR amplicons were prepped with DNA loading dye and run on an 8% (bis)acrylamide/TBE gel. Visualization was performed on a Molecular Imager^®^ Gel DocTM XR+ System (Bio-Rad, Hercules, CA, USA) supported by the Image Lab software (Bio-Rad, Hercules, CA, USA). Original blots can be found online in the Appendix A containing Appendix A.

### 4.4. Irradiation Procedure

At 24 h prior transplantation, animals (early froglet stage) were sub-lethally irradiated up to 12 Gy with X-rays. Froglets were placed individually in 50 mL Falcon tubes filled with 25 mL filter sterilized frog water and irradiated in a XRAD320 device (Precision X-Ray, Inc., North Branford, CT, USA) at approximately 120 cGy/min. 

### 4.5. Tumor Cell Transplantation

Tumor cell suspensions were prepared manually by dissecting tumor pieces and subsequently washing them with sterile amphibian phosphate buffered saline (APBS), after which they were poured through a 40 µm cell strainer (Falcon^TM^) using tweezers to mince the tumor and APBS for flushing. An aliquoted 20 µL of cells was mixed with 180 µL 0.1% trypan blue solution to count the density of living cells. Subsequently, the tumor cell suspension was centrifuged for 5 min at 240× *g* (RT) and resuspended with APBS to the appropriate concentration. Recipient host frogs (*rag2^−/−^*, irradiated or WT) were sedated using a 2 g/L MS222 (Tricaine methanesulfonate) solution diluted in sterilized frog water adjusted to pH 7 using sodium bicarbonate. Each recipient host animal was injected intraperitoneally with a 100 µL tumor cell suspension (ranging from 3.3 × 10^6^ to 2 × 10^7^ tumor cells) using BD Micro-Fine Demi 0.3 mL 0.3 mm (30G) × 8 mm syringes. Post-transplantation, injected animals were housed separately and monitored closely for any signs of engraftment or discomfort. For all animal experiments, ethical approval was obtained, and guidelines set out by the ethical committee were followed.

### 4.6. Blood Counts

Peripheral blood or intraperitoneal fluid was isolated by cardiac puncture or intraperitoneal (IP) lavage, respectively. For the IP lavage, a small incision was made in the skin of the belly and the abdominal muscle wall after which 100 µL APBS was used for rinsing the IP cavity. Approximately 10 µL IP fluid diluted in APBS was collected for further processing. Immediately after collection, cells were diluted 1:50 in Natt and Herrick reagent, a methyl violet based staining solution, for downstream counting and histological analysis [29,30]. Counts were performed using a Bürker hemocytometer (Marienfeld, Lauda-Königshofen, Germany). For each Natt and Herrick sample at least 2 × 6 regions were counted (minimum 150 cells per count). 

### 4.7. Imaging, Histology and Immunohistochemistry 

Animals were euthanized by lethal incubation in a Benzocaine solution (500 mg/L) until heart beating stopped. Macroscopic images were taken with a Carl Zeiss StereoLUMAR.V12 stereomicroscope. Dissected organs or tumors were fixed overnight in 4% PFA at 4 °C and subsequently dehydrated and paraffinized. Organ slices (5 µm) were generated by microtomy and stained with hematoxylin and eosin using the Varistain™ 24-4 Automatic Slide Stainer (Thermo-Scientific, Waltham, MA, USA) for classical histological assessment. For immunohistochemistry (IHC) experiments, paraffin sections were dewaxed and pressure cooked using citrate buffer (10 mM citric acid, 0.05% tween-20, pH 6) for antigen retrieval, and 3% natural goat serum was used for blocking. The following primary antibodies were used: IgG anti-human CD3 antibody (1:200, clone CD3-12, Bio-Rad, Hercules, CA, USA) and anti-PCNA antibody (1:1000, PC10, Dako, Santa Clara, CA, USA). The following secondary antibodies (all 1:500) were used: Biotinylated Goat Anti-Rat Ig (559286, BD Pharmingen, Franklin Lakes, NJ, USA) and Biotinylated Goat Anti-Mouse Ig (E0433, Dako, Santa Clara, CA, USA). DAB was used as the chromogenic method of detection and the signal was developed using the VECTASTAIN Elite ABC HRP Kit (PK-6100; Vector Laboratories, Newark, CA, USA) combined with ImmPACT DAB Peroxidase (SK-4105; Vector Laboratories, Newark, CA, USA). Finally, samples were counterstained with hematoxylin. All IHC experiments included ‘no primary antibody’ controls (data not shown). Imaging of sections was performed by using an Olympus BX51 Discussion Microscope. For quantification of the CD3 stained slides, the QuPath software tool [20] was used. Slides were acquired using the ZEISS Axioscan 7 machine at 20× magnification with a resolution of 0.22 μm/pixel.

### 4.8. Statistical Analysis

Comparisons and conclusions between the experimental and the wild-type groups were statistically supported by two-sided Student’s t-tests (non-significant *p* ≥ 0.05, * *p* < 0.05, ** *p* < 0.01, *** *p* < 0.001, **** *p* < 0.0001). Bar charts shown represent means with SD as error bar.

## 5. Conclusions

In summary, we document the generation of a *rag2* mutant *Xenopus tropicalis* line that allows the stable engraftment of tumor cells. We showed the successful allotransplantation in this line with both a thymic tumor originating from an adult *tp53*^−/−^ animal and liver tumor cells from an *apc^+/−^/tp53^−/−^* adult frog. Furthermore, we optimized a protocol for the generation of immunocompromised *X. tropicalis* froglets by using sub-lethal X-ray irradiation. We could show that irradiation up to 12 Gy achieved the most efficient suppression of the host immune cells. Finally, by means of successful transplantation experiments, we could also demonstrate the utility of these immunocompromised animals. In conclusion, we believe that the availability of these novel tools will greatly enhance experimentations in *X. tropicalis* for studying human cancer.

## Figures and Tables

**Figure 1 cancers-14-04560-f001:**
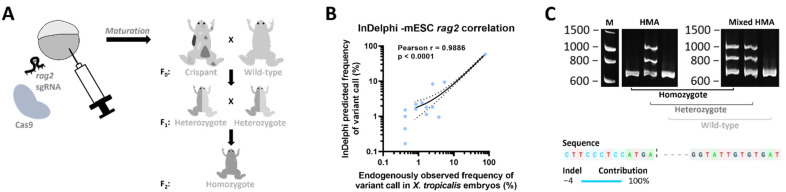
Generation of the *X. tropicalis rag2^−/−^* knockout line. (**A**) Embryos were injected with an sgRNA targeting the *rag2* gene along with Cas9 protein. When sexually mature, animals were outcrossed to wild-types to obtain heterozygous animals that were subsequently incrossed to obtain *rag2* homozygous mutant animals in the F_2_ generation. (**B**) Scatter plot showing correlation between in vivo observed mutational CRISPR repair outcomes in injected embryos (*x*-axis) versus predicted outcomes using the inDelphi algorithm tool (*y*-axis). Dashed lines show the 95% confidence interval corresponding to the best-fit linear regression line (solid line). (**C**) Genotyping of sampled F_2_ animals. Images taken from DNA electrophoresis gels after performing a normal HMA (left) and mixed HMA (right). Normal HMA included heating of the sampled PCR amplicons followed by slowly cooling and loading on the gel, while, for mixed HMA, sampled PCR samples were first mixed with wild-type *rag2* amplicons, after which the HMA was performed. Multiple bands present in both gels indicate heterozygous animals, while extra bands only appearing after performing the mixed HMA (right gel) relate to homozygous mutant animals. Absence of any extra bands is indicative of wild-type animals. Presence of a 4 bp deletion in homozygous mutant animals was confirmed by Sanger sequencing.

**Figure 2 cancers-14-04560-f002:**
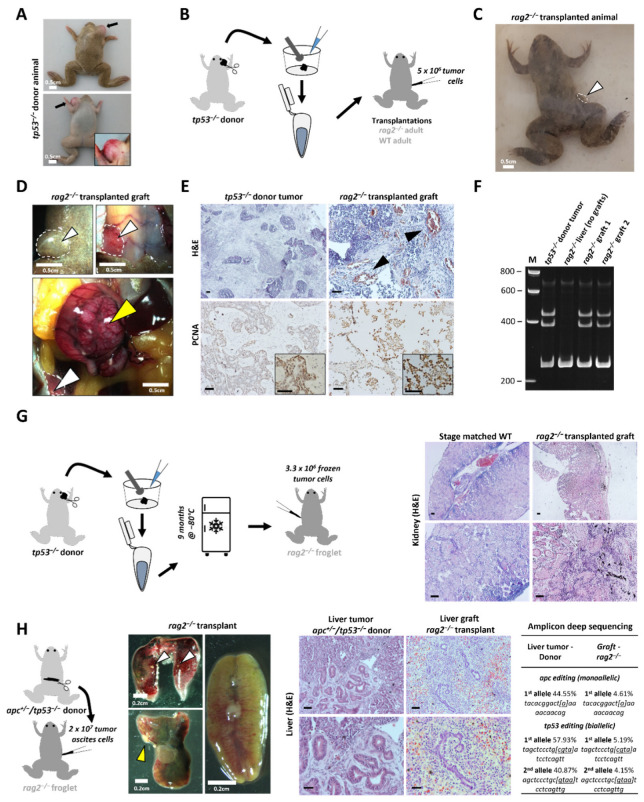
Validation of allografting in *X. tropicalis rag2^−/−^* animals. (**A**) *tp53^−/−^* donor animal harboring a thymic tumor (black arrows). (**B**) Transplantation strategy including the generation of a cell suspension using a 40 µm strainer followed by IP injections in a *rag2*^−/−^ adult and a wild-type adult control (both 5 × 10^6^ live cells). (**C**) A *rag2*^−/−^ transplanted animal with a subcutaneous outgrowth close to the injection site (white arrow, white dashed line) 10 weeks post-transplantation. (**D**) Microscopy images (ventral view) of *rag2^−/−^* transplanted animal showing the engrafted tumor at the injection site before and after removal of the skin (top panels) and internal (top right and bottom). The tumor is also visible upon opening of the abdominal cavity (white arrowheads, white dashed line), in addition to a large tumor mass associated with the intestinal mesenterium (yellow arrowhead). (**E**) H and E and IHC stained sections from the primary tumor in the *tp53*^−/−^ donor animal and the tumor graft in the transplanted *rag2*^−/−^ animal. Black arrowheads indicate regions with neovascularization. (**F**) Mixed HMA analysis for the *tp53* gene on DNA from *tp53*^−/−^ tumor sample (donor animal), liver (without grafts) and two tumor grafts obtained from the transplanted *rag2*^−/−^ animal. (**G**) Transplantation experiment starting from frozen *tp53^−/−^* tumor cells. H and E images from the *rag2^−/−^* transplant kidney versus a stage matched wild-type are shown. (**H**) Transplantation experiment starting from tumor ascites cells from an *apc^+/−^/tp53^−/−^* liver tumor bearing animal. Dissection micrographs from kidney, liver and lungs of *rag2^−/−^* transplant. Yellow and white arrowheads show visible engraftment regions in liver and lungs, respectively (left). H and E images from the liver tumor in donor animal and liver grafts in the transplanted animal (middle). The presence of tumor cells in the liver of the transplanted *rag2^−/−^* animal is confirmed by amplicon deep sequencing at the *apc* and *tp53* target site (right). All black scale bars are 50 µm. White scale bars are 2 mm.

**Figure 3 cancers-14-04560-f003:**
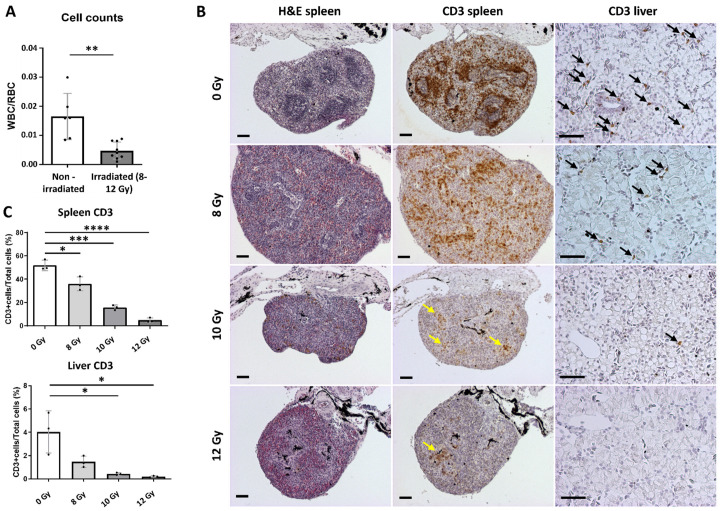
Allografting in irradiated wild-type *X. tropicalis* animals. (**A**) Plots showing hemocytometer cell counts as represented by white blood cell (WBC)/red blood cell (RBC) ratios of irradiated animals and non-irradiated controls. (**B**) H and E and anti-CD3 immunostained sections from spleens and livers of all 4 groups. Yellow arrows show CD3 positive zones in the spleen; black arrows show CD3 positive cells in the liver. (**C**) IHC quantified CD3 data of spleens and livers using the open source digital analysis tool QuPath [20]. * *p* < 0.05, ** *p* < 0.01, *** *p* < 0.001, **** *p* < 0.0001. All scale bars are 50 µm. Bar charts shown represent means with SD as error bar.

**Figure 4 cancers-14-04560-f004:**
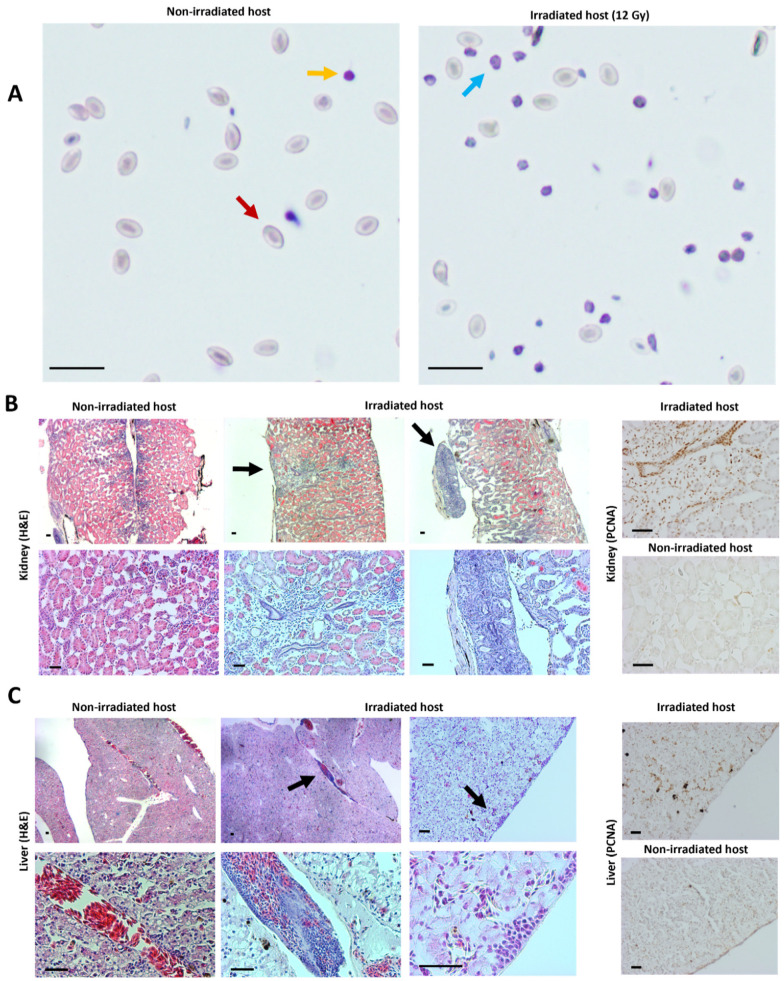
Engraftment of *tp53^−/−^* tumor cells in irradiated wild-type *X. tropicalis* froglet. (**A**) IP fluid from tumor cell transplanted irradiated and non-irradiated control animals, stained with Natt and Herrick reagent. Red and yellow arrows indicate an RBC and a lymphocyte, respectively. The blue arrow shows a tumor blast cell. (**B**) H and E- (left) and PCNA-stained (right) sections of engrafted regions in kidney (black arrows) from transplanted irradiated froglet compared with respective kidney sections in the transplanted non-irradiated control froglet. (**C**) H and E- (left) and PCNA-stained (right) sections of engrafted regions in liver (black arrows) from transplanted irradiated froglet compared with respective liver sections in the transplanted non-irradiated control froglet. All scale bars are 50 µm.

## Data Availability

The data presented in this study are available in this article and the Appendix A.

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
