# Peer review of "Engraftment of Allotransplanted Tumor Cells in Adult rag2 Mutant Xenopus tropicalis"

_cancers, 2022, doi:10.3390/cancers14194560_

Round 1
Reviewer 1 Report
The manuscript by Tulkens et al. described successful allografting of tumor cells in Xenopus tropicalis via the establishment of a rag2 knockout frog line as well as optimizing X-ray irradiation of wild-type Xenopus tropicalis froglets. The manuscript is well written.
The following issues may be addressed to improve the manuscript.
1. It would be nice to provide hematopoietic cell quantification evidence clarifying if the rag2-/- frogs generated in this study have zero or only reduced numbers of functional T and B cells.
2. The cell types in Figure 4A should be clearly labelled. Relevant control (PCNA staining of kidney and liver samples from non-irradiated host) should be provided for Figure 4C.
3. Based on the information described in the Results and Materials and Methods, the irradiation was carried out with X-ray. The “gamma irradiation” description in the Abstract should be corrected.
Author Response
We thank the reviewer for his/her appreciation of our manuscript and the suggested changes to improve the manuscript.
1. It would be nice to provide hematopoietic cell quantification evidence clarifying if the rag2-/- frogs generated in this study have zero or only reduced numbers of functional T and B cells.
Response: Unfortunately there are no suitable antibodies available for reliably evaluating T- and B-cells in Xenopus, and definitely not for interpreting their differentiation status. Hence we can only rely on the functional phenotyping (i.e. the acceptance of tumor grafts).
2. The cell types in Figure 4A should be clearly labelled. Relevant control (PCNA staining of kidney and liver samples from non-irradiated host) should be provided for Figure 4C.
Response: we thank the reviewer for these suggestions and have added them to the revised manuscript.
3. Based on the information described in the Results and Materials and Methods, the irradiation was carried out with X-ray. The “gamma irradiation” description in the Abstract should be corrected.
Response: we thank the reviewer for pointing out this mistake, which has bene corrected in the abstract.
Reviewer 2 Report
This is an excellent paper clearly written and documented by tha ample of well prepared figures should be accepted right away
Authors described the generation of a rag2-/- knockout line in Xenopus tropicalis that is deficient in functional T and B cells. They validated the line by allografting experiments with primary tp53-/- and apc+/-/ tp53-/- donor tumors. Also, they optimized available protocols for sub-lethal gamma irradiation of wild type X. tropicalis froglets. Irradiated froglets were used for transient engraftment of transplanted X. tropicalis tumor cells. These novel rag2-/- line and the irradiated froglets will expand the use of the diploid amphibian X. tropicalis as a model for studying human cancer. The topic is very original. It is extremely rare that a nonmammalian, and especially aquatic animals are uses as a model for human diseases. In contrast to Xneopus laevis, the Xenopus tropicalis is a diploid species, which makes is especially useful for studying genetic mutations relevant to human cancer. This is a unique and carefully designed study which introduces X. tropicalis as a human cancer model system I do not recommend any improvements at methodology part. As all studies, the results create additional and new questions, which can be addressed by further studies in the field authors described the generation of a novel X. tropicalis rag2-/- line. Such line is extremely useful for transplantation experiments because these animals lack mature T and B cells, so partially mimic the nude mouse phenotype. These animals can be used for intraperitoneal injection (grafting) of tumor cells the reference list is extensive and includes all publications important for the subject The manuscript is illustrated by the excellent diagrams of the experimental protocols, and the results are supported by many excellent quality microscope images of tissue sections
Author Response
We greatly appreciate the positive comments by this reviewer. No changes or additions were requested.